# Strain Release Behaviour during Crack Growth of a Polymeric Beam under Elastic Loads for Self-Healing

**DOI:** 10.3390/polym14153102

**Published:** 2022-07-30

**Authors:** Mohammed Dukhi Almutairi, Sultan Saleh Alnahdi, Muhammad A. Khan

**Affiliations:** 1School of Aerospace, Transport, and Manufacturing, Cranfield University, Cranfield MK43 0AL, UK; s.alnahdi@cranfield.ac.uk; 2Centre for Life-Cycle Engineering and Management, Cranfield University, College Road, Cranfield MK43 0AL, UK; 3Sustainable Manufacturing Systems Centre, Cranfield University, College Road, Cranfield MK43 0AL, UK

**Keywords:** 3D printing, ABS simple beam, TPU origami capsule, embedded structure, self-healing mechanism, double cantilever beam test

## Abstract

The response of polymeric beams made of Acrylonitrile butadiene styrene (ABS) and thermoplastic polyurethane (TPU) in the form of 3D printed beams is investigated to test their elastic and plastic responses under different bending loads. Two types of 3D printed beams were designed to test their elastic and plastic responses under different bending loads. These responses were used to develop an origami capsule-based novel self-healing mechanism that can be triggered by crack propagation due to strain release in a structure. Origami capsules of TPU in the form of a cross with four small beams, either folded or elastically deformed, were embedded in a simple ABS beam. Crack propagation in the ABS beam released the strain, and the TPU capsule unfolded with the arms of the cross in the direction of the crack path, and this increased the crack resistance of the ABS beam. This increase in the crack resistance was validated in a delamination test of a double cantilever specimen under quasi-static load conditions. Repeated test results demonstrated the effect of self-healing on structural crack growth. The results show the potential of the proposed self-healing mechanism as a novel contribution to existing practices which are primarily based on external healing agents.

## 1. Introduction

Three-dimensional (3D) printing or additive manufacturing (AM) of smart polymers is a rapidly expanding area of technology. The variety of AM techniques available suggests it may be possible to flexibly manufacture smart but costly materials with minimum waste. On-demand or autonomous repair of forms of damage, such as cracks or scratches, can increase the operational life of products and can be facilitated using man-made polymers which are autogenous or intrinsically self-healing. A balance between healing and strong mechanical properties can be achieved by designing the architecture of the polymer to incorporate dynamic or reversible bonds [1,2,3,4,5,6]. A great deal of work still needs to be performed to successfully implement self-healing mechanisms in real applications, with most previous studies of self-curing structural damage having taken place only at a laboratory scale. The majority of reported mechanisms have been based on external disturbances such as heat-generated cracks or a chemical reaction triggering the healing mechanism within the structure. Existing mechanisms tend to depend on some form of external interference and, most of the time, work only for more significant damage. Consequently, it is virtually impossible to implement current self-healing mechanisms such as those in 3D printed products whilst they are functioning; this is a particularly important consideration in some vital applications [7,8,9].

An alternative approach to creating smart 3D printed products is to embed novel origami-inspired capsules into the layers of a printed component. For essential applications, in particular, such capsules could create an artificial hormone network that would make 3D printed products safer and considerably more dependable [10,11,12]. Standard fused deposition modelling can be utilised to embed these capsules when printing the required component, which would be a cost-effective solution for large-scale production. This is somewhat similar to the manner in which the human hormone system actuates when a virus or bacteria enters the body. The use of a strain removal-based actuation via origami-inspired capsules could radically transform the self-healing capacity within components or structures. Strain removal from an entire component could thus be initiated by any form of surface or subsurface damage. For strain removal to take place at a sub-surface level, the capsules could unfold and expand [13,14].

However, the actuation or unfolding of such capsules under strain release due to crack initiation or growth within a structure requires an understanding of its mechanical behaviour, especially in embedded conditions. To introduce and control such a process requires a workable relationship between the initial stress on the embedded capsules, the displacement of the origami folded parts in a direction to release the strain, and the magnitude of the strain released. An overall understanding of the mechanical behaviour of any selected polymer under elastic and plastic loads is necessary to assess its usefulness in the form of an origami capsule to provide the necessary strain release control.

The mechanical behaviour of ABS polymer components has been investigated for many years, and the basic features, such as stress–strain curves, are adequately known [15,16]. Such behaviour is measured elastically for very small strains and slightly larger strains when overcoming the intermolecular barriers to segmental rearrangements [17]. However, the complex properties of ABS polymer materials are temperature dependent, which has driven further investigation to determine what relationships exist between strain, stress, and temperature [18,19].

Previous research into the mechanical properties of thermoplastic polyurethane (TPU) and thermoset acrylonitrile butadiene styrene (ABS) provides help in understanding the dynamics of such beams under load. Yuan et al. investigated the behaviour of a graded origami structure under quasi-static compression. A beam was fabricated using ABS material with flat brass sheets, 0.3 mm thick, implanted between the moulds, which were then compressed. Results indicated that the proposed origami structure showed plane stiffness and higher energy absorption to external loads [20]. However, the work lacked analysis in terms of geometric optimisation and behaviour under impact load. Hernandez et al. presented a kinematic study of origami structures for both elastic and plastic polymeric beams. After assessing various design structures, it was found that the kinematic variables of the structural model could fully explain the configuration of elastic origami structures within the beam [21,22]. However, the model developed by the researcher is far more complete and needs fewer variables for efficient FEA. Li and You researched open section origami beams to demonstrate energy absorption. Their research focused on designing a beam which included origami geometries and which retained its cross-sectional height better than conventional beams when subject to large externally imposed bending deformations. Despite numerical simulation, the model did not develop origami geometries able to cope with symmetrical vertical loads; also, the energy absorption model needed to be validated [17]. Nevertheless, origami-based encapsulation has shown promising results [23], but tests of mechanical strength and healing properties tend to have been carried out on soft and weak materials [24].

The encapsulation of folded material, such as TPU in rigid and static structures, can induce self-healing properties in a structure, assisting it in overcoming extreme fatigue conditions, material degradation, and failure due to micro-cracks [25,26]. Moreover, by activating the self-healing process, the material becomes safer and more durable, saves the time and cost of replacing particular items, and reduces inefficiencies incurred due to damage [27,28].

The four-point flexure response of the ABS beam has been researched by Dhaliwal and Dundar and showed high impact resistance and toughness. Their work examined the strain rate using the Generalised Incremental Stress-State Model. Though the compressive elastic modulus of ABS is found to be much higher than its tensile elastic modulus, the Von-Mises is yielded at a much lower force [29]. This means that at higher deformations, the ABS beam may not produce the predicted theoretical results. Therefore, it is necessary to continue to research self-healing techniques of polymers using origami structures.

Lee [30,31] conducted an experiment using a large elastic bending machine to investigate the elastic energy behaviour of curved–creased origami to assess material bending behaviour. As the first step, an origami design model was developed to use different folds to produce the patterns necessary to make the 3D form required to meet a prescribed buckling criterion. The model was then used to simulate the shape of the origami capsule required, after which the results could be experimentally validated. The study by Lee [30] showed that skewed curved–creased laminated surfaces could help in assembling compliant and energy-absorbing structures, but the study itself did not provide any direct evidence for using this mechanism for self-healing. However, once the results were validated, it allowed a healing process using origami capsules to be simulated.

In this work, the response of polymeric beams of ABS and TPU materials under elastic and plastic loads is investigated. The experimentation process included the use of strain gauges of different thicknesses to determine the deflection of the cantilever beam under test [32,33]. The tests included observation of the effects of the material and binder on two types of 3D-printed beams and were designed to test their elastic and plastic responses under different bending loads. These responses were used to develop an origami capsule-based novel self-healing mechanism triggered by crack propagation due to strain release in a structure.

The origami capsules were cross-shaped and made of four small beams that could be folded or elastic deformed and embedded in the main beam structure. Under the strain released due to crack propagation in the main beam, the small beams of the origami capsule unfolded in the direction of the path of the crack and hence increased the structure’s resistance to crack propagation. This increase in the crack resistance was validated in a delamination test of a double cantilever specimen under quasi-static load conditions. Repeated results demonstrated the effect of self-healing on structural strength against crack growth. The results show the potential of a proposed self-healing mechanism as a novel contribution to existing practices, which are primarily based on external healing agents

The paper is structured as follows: Section 2 describes the methodology, including the selection of materials, experimental setup and procedure of the simple beam, origami beam and origami embedded structure. Section 3 provides the results and discussion. The conclusions are presented in Section 4.

## 2. Methodology

In this section, various techniques used to prepare and characterise the samples are described. Specifically, bending loads were placed on the end of rectangular beams of the polymeric materials to gain a better understanding of their elastic and plastic behaviour. The four steps in this research are shown in Figure 1. The first step was selection of the polymeric material and included the preparation of the specimen and experimental methods. The second step was design of the polymeric structure, including the origami capsule. The third step was the design of the experiment to investigate the properties of the samples, including bending moment and delamination tests. In the final step, the tensile test machine was used to obtain strain–stress curves, bending points and delamination effects using a single bending moment, see Figure 1.

### 2.1. Material Selection

The first polymeric material selected was ABS, one of the most common raw materials used for printing beams via fused deposition modelling. ABS has good impact resistance, high rigidity, strain resistance, etc., even at low temperatures [34], properties that make it a suitable material for the intended application. Sample parts were fabricated at variable parameters and tested for bending strength. The second material chosen was TPU. This is of interest because of its versatility in terms of a wide range of mechanical properties, good abrasion resistance and low density. TPU is more elastic than ABS and very suitable to be folded as capsules. TPU has additional benefits compared to other polymers, such as being extremely flexible, durable and smooth to the touch.

A Raise3D Pro printer was used to print the beam-based origami capsule and embedded structure beam. The 3D printed samples and capsule were produced with two printing parameters: orientation and layer thickness. The platform was heated to 80 °C with a screw speed of 50 mm/s. At least 1 kg of filaments with a diameter of 1.75 ± 0.05 mm served as the extender. During the printing process, the slicer programme used this diameter to calculate the required feed rate [35,36,37]. The mechanical printer parameters are presented in Table 1 and depicted in Figure 2 and Figure 3.

### 2.2. Specimen Preparation

The specimen was designed as a simple beam with embedded structure. The design of the embedded capsule is shown in Figure 4, which also shows its dimensions. The cantilever beam was designed using the inventor software, as shown in Figure 4a, sample thickness is 3.0 mm, length 145 mm, and width is 10 mm. Figure 4b the origami capsule thickness 3.0 mm, length 19 mm, and width 5 mm. Figure 4c shows the length of specimen, 193.0 mm, width 30 mm, and thickness 5 mm. These dimensions were maintained in all tests.

G-code files for printing the above specimens on a 3D printer were created using Idea Maker software (Raise3D pro2). The 3D printer process from drawing to fused deposition is shown in Figure 5.

At least three samples were printed of each simple beam, capsule, and embedded structure. The infill density was 100% in all cases. First, TPU and then blends containing 5, 10, and 20 wt% TPU were printed. Nozzle temperature was set to 60 °C for all capsules. Printing speed was 40 mm/s, and print bed temperature was 60oC. Similarly, ABS simple beams were also printed with infill densities of 40, 60, and 80 wt%, respectively. Here the printing speed was constant 60 mm/s with nozzle diameter 0.4 mm. The print bed temperature was 80 °C and 100 °C. For each configuration, two samples were printed [8].

### 2.3. Design of Experiment

In this experimental study bending load and delamination tests were carried out. Both sets of experiments began with the printing of samples, the simple beams, the origami capsules and the origami capsules embedded in the beams. The samples were then subjected to bending load and delamination tests using an Instron 5944 Universal Testing Machine (UTM). Specifically, the bending load was applied to better understand the elastic and plastic behaviour of ABS and TPU. The stresses were calculated according to the force provided by the UTM. In addition, video images taken during the loading determined the deflection of the beam at 15 different points along its length. This provided the overall response of the beam under bending load. Furthermore, each quasi-static double cantilever beam (DCB) test was conducted three times using the UTM. The loading value was measured with a load cell attached to the tensile test machine. The opening displacement and crack length were measured with a camera.

#### 2.3.1. Simple Beam and Origami Beam

In this work, the parameters were set as shown in Table 2 and Table 3 using design of experiment methodology. The ABS simple beam and TPU origami capsule were manufactured. Sensor calibration was performed (3 times for each beam thickness: 0.5 mm, 1.0 mm, 2.0 mm, and 3.0 mm. Deformation load and deflection data were recorded, and the data (strain, applied load, and deflection) plotted using Excel.

**Table 2 polymers-14-03102-t002:** Experimental setup for simple beam.

Sample Number	Beam Thickness mm	Dimensions, mm (Length/Width)	Loads (g) Attached to the Beam, See Figure 6
1	0.5	145/10	1, 2, 3, 4, 5, 6, 7, 8, 9, 10
2	1.0	145/10	1, 3, 5, 10, 15, 25, 35, 45, 55, 75, 100
3	2.0	145/10	5, 10, 20, 30, 50, 70, 90, 110, 160, 210, 310, 410
4	3.0	145/10	5, 10, 20, 30, 50, 70, 90, 110, 160, 210, 310, 410

**Figure 6 polymers-14-03102-f006:**
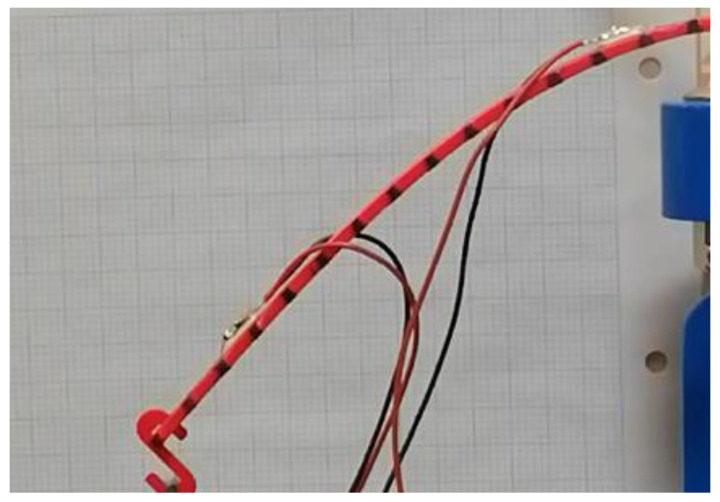
Deflection points.

#### 2.3.2. Origami Beam Embedded Inside Structure of the Beam

A simple experiment was conducted to record load and displacement using a delamination test on the specimen beams printed using ABS polymer, as shown in Table 4 The stress/strain relationship with and without the origami capsule was then evaluated.

The initial hypothesis was that specimens containing the origami structures would be more resilient and exhibit higher tensile strength when loaded axially. Conversely, specimens that did not have origami structures embedded within them should exhibit lower resilience or lower tensile strength. In order to test this hypothesis, two hollow 3D beam samples were printed that could be joined later by mechanical means. One of the samples contained 3D printed origami structures embedded inside using an adhesive. The other, the control, was the same 3D printed beam but without the origami structure embedded within it. The specimens were loaded and pulled (tensile loading) axially. Force vs. displacement (F/D) curves were obtained, which corresponded to the stress/strain relationship. In order to convert F/D curves into a stress/strain relationship, force values were divided by the cross-sectional area of the beam, while D values were divided by the initial gauge length. A schematic of the experimental procedure is presented in Table 5.

### 2.4. Experiment Setup and Procedure

In this experiment, a micrometre was used to apply a deflection to the end of a beam. Before starting the experiment, the dimensions of our simple beam and origami beam were measured using inventor software. The dimensions of the beams are given in Table 2 and Table 3 above.

For each specimen, the following set of procedures was carried out.

First, prepare the surface of the test piece by applying conditioner and neutralisers. A step-by-step procedure was developed. The process of bonding the strain gauge should be carried out precisely without errors. Notably, the surface area of the strain gauge should be stuck together by first cleaning the surface with sandpaper and then using conditioners to neutralise the free-end and the fixed support. Finally, to complete the surface preparation of the beam, a generous volume of the neutraliser is applied and wiped out with the cotton ball.To further explain the process for educational purposes, the bonding area must be cleared with alcohol/acetone. After clearing the surface, the necessary marks are placed on the bonding site, preferably with a fine graphite pencil, such that no residual deposition affect the measurement.Clamping of the beam: the flat portion of the ABS beam was clamped in the test machine.Place the strain gauges on the sample. One in free end and other one close to the fixed support, as shown in Figure 7.

**Figure 7 polymers-14-03102-f007:**
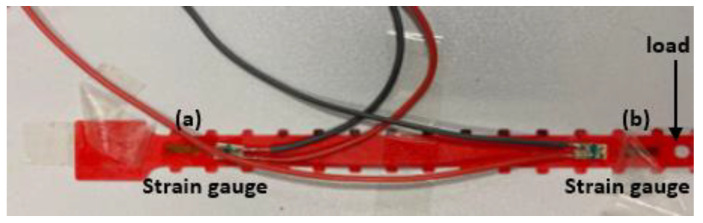
Plane view of strain gauge mounting points on test beam (**a**) fixed support and (**b**) free end.

Connect the strain gauges to the DAQ-card and the Signal-Express softwareCalibrate the strain gauges with no load on the sample and set readings to zero.Measure the distance between free end of each sample and the nearest strain gauge.Apply loads progressively from 0 N to 4.02 N and measure the corresponding strain on each of the two strain gauges. Remove the masses in the reverse order in which they were added to produce a hysteresis plot.Placing the protractor parallel to the edge of the clamping machine (i.e., starting point of the beam). The fixing should be firm, so there is no unwanted movement of the protractor.Camera orientation: The camera was placed parallel to the longitudinal side of the beam such that the protractor could be easily seen. The distance between the camera and the beam was 30 cm.

#### 2.4.1. ABS Simple Beam Behaviour in Normal Condition

A simple ABS beam of 145 mm length, see Figure 4, was fixed at one end as cantilever beam. HD camera was chosen for strain measurement rather than crosshead displacement because of the compliance of the loading mechanism and load cells, which is typical in such tests. HD Camera deflection measurement on both sides of the sample compensates for any lateral bending during loading. The procedure for testing the sample begins by setting the selected beam thickness. Next, the loads were applied at the free end and readings of the deflections taken. For every deformation, a picture of each point was taken, as shown in Figure 6. Specimen preparation only required a light-ordered pattern of black paint on beam, see Figure 6 on the white plastic background. Images of the samples were captured via camera, and deflection data obtained. After a sample was placed in the testing machine and a preload applied, a pair of reference images (one image per camera) were taken of each side of the sample. The applied loads ranged from 1 g to 410 g, depending on beam thickness, see Table 2. These were placed at the end of the beam. The wires used to connect the strain gauge to the DAQ (NI 9235) [38]. The D-card meter was connected to the computer via chassis (NI cDAQ-9174) for post-processing and data analysis.

This methodology proved to be efficient, and testing of a single specimen could be performed in matter of minutes, including mounting the specimen, taking initial undeflected images, and loading the specimen through to failure. The complete setup, including the camera and universal testing machine used for bending load.

#### 2.4.2. Polymeric Origami Beam Behaviour

The simple ABS beam was replaced by one with an origami insert; first, a “cross”, see Table 3. The loads were applied to the origami capsule, and measurements taken via the computer using the signal conditioning unit and data logger. The experiment was carried out with three tubes of thicknesses of 1.0, 2.0, and 3.0 mm. Each time the camera was set to a required value, and the corresponding strain values were recorded. After repeating the experiment three times, the average value of the results was obtained and noted. The origami capsules were designed using inventor software with different shapes to test the workability of different capsules, Figure 8. The designs of the capsules were such that their geometrical features were confined under the initial pre-stressed conditions.

#### 2.4.3. Beam Behaviour with the Origami Capsule Contained within It

The setup and experimental design for the DCB test are shown in Figure 9. The beams were, as shown in Figure 4c, 30 mm wide and 193.0 mm long, with a 40 mm longitudinal pre-crack extending from the front of the specimen, see Figure 9. End tabs of 30 mm width were glued on the external faces of the specimens on either side of the pre-crack and pinned to an electromechanical uniaxial testing machine with a 500 N load cell. The DCB tests were performed on an Instron testing machine with displacement rates that could be varied between 0.05 to 0.10 mm/s

Because the ABS samples were transparent, crack length was directly recorded from the top using a camera. Each sample was tested 3 to 5 times. The corresponding energy release rates and critical energy release rates were calculated using a simple beam. The programmed loading history was not monotonic: indeed, eight loading and unloading cycles at the same displacement rate were programmed into the machine to verify the absence of permanent deformations, which would indicate parasite sources of energy dissipation. For each cycle, the maximum displacement at loading was defined, as well as a minimum force at unloading, set at 5 N to avoid compression of the test specimen.

Three different types of specimens were printed, one for each set of parameters, 1 mm, 2 mm, and 3 mm thickness. Three specimens were manufactured and tested under the same conditions for each thickness to confirm the experimental repeatability of the results obtained.

Mode I interlaminar toughness tests were performed on DCB beams, see Figure 4c containing an origami capsule orientated normal to the direction of crack growth [39,40,41,42,43]. The DCB specimens had a 40 mm long pre-crack at the front of the specimen, as shown in Figure 9. Two hinges were glued onto the top and bottom surfaces of the sample so they could hold the ends of the arms of the DCB specimen. The delamination crack growth in the direction of the origami capsule was as shown in Figure 9.

## 3. Results and Discussion

### 3.1. Results for ABS Simple Beam Behaviour

The displacement responses of the ABS materials with bending loads applied at the tip are shown in Figure 10. It is evident that for the 3.0 mm thick beam within the elastic limit, the maximum stress yielded a deflection of 56 mm. However, for the 0.5 mm thick beam, observed a deflection of 79 mm. Within the elastic limit, for small deflections, the value of the stress is directly proportional to the force and inversely proportional to the thickness:(1)$$Deflection\;\propto \frac{f\left(force\right)}{f\left(\;thickness\right)}$$

The values of force and maximum deflection were used to calculate the strain energy of the beam. It was assumed that the amount of stress applied is wholly converted into strain energy, which is represented as:(2)$$Maximum\;Strain\;energy\;of\;beam\;\left(U\right)=\frac{\sigma^{2}}{2E}\;Ba$$
where σ represents the stress applied, E is the elastic modulus of the material, B is the beam’s thickness, and *a* indicates the length of the beam.

From the force–deflection curves shown in Figure 10, it is seen that the maximum deflection for the beam of 3.0 mm thickness was 64 mm at the maximum applied force of 4.022 N. The maximum deflection for the beam of 2.0 mm thickness was 78 mm, and the maximum applied force was again 4.022 N. The maximum deflection for the beams of 1.0 mm and 0.5 mm thicknesses was 93 mm at the maximum applied forces of 1.73 N and 0.55 N, respectively. Hence it follows that:

Maximum strain energy beam at 0.5 mm beam thickness
$$=\frac{191.48^{2}}{2\;\times \;\left(1681\right)}\;\times \;5\;\times \;145=7907\;N.\mathrm{mm}$$

Maximum strain energy at 1 mm beam thickness
$$=\frac{98.19^{2}}{2\;\times \;\left(1681\right)}\;\times \;10\;\times \;145=4158\;N.\mathrm{mm}$$

Maximum strain energy at 2 mm beam thickness
$$=\frac{87.48^{2}}{2\;\times \;\left(1681\right)}\;\times \;20\;\times \;145=6601\;N.\mathrm{mm}$$

Maximum strain energy at 3 mm beam thickness
$$=\frac{38.88^{2}}{2\;\times \;\left(1681\right)}\;\times \;30\;\times \;145=1956\;N.\mathrm{mm}$$

From the calculated values, it is noted that maximum strain energy is observed for the 0.5 mm thick beam, 7907 N.mm and the least value of strain energy is observed for the thickest beam, 3.0 mm, which is 1956 N.mm. This validates the findings that maximum force and beam thickness yield minimum strain energy, and the lower the magnitude of force and beam thickness, the higher the strain energy.

In Figure 10, the vertical black lines indicate the initiation of the plastic regime of each beam; it is clear that the greater the thickness of the beam, the greater the force required. This is the reason why the plastic region for the 0.5 mm beam commenced at 0.15 N, whereas for the 3.0 mm beam, the force required was 3.04 N.

With the observed changes for the maximum deflection with respect to different loads, there is a need to analyse whether or not the values change with position. This was performed by plotting a 3D surface graph in the next section (see Figure 11) to show the parametric relationship between force, deflection, and position of the beam.

It is evident that the gradient increases in value as the thickness of the beam increases in four stages, from 0.5 mm to 2.0 mm.

With respect to different positions of the beam, we have assumed that deflection at different forces varies accordingly.

The curves shown in Figure 12 demonstrate strain energy as a function of applied force for simple ABS beams of (a) 0.5 mm, (b) 1.0 mm, (c) 2.0 mm, and (d) 3.0 mm thickness. The trend indicates that the strain energy attained its maximum value for 0.5 mm thickness (7907 N.mm), followed by 6600 N.mm for 2.0 mm, 4158 N.mm for 1 mm, and 1955 N.mm for 3 mm. It was also noted that the strain energy for all thicknesses except 2.0 mm had reached zero before 0.5 N, while the strain energy for the 2.0 mm thick beam reached zero value only as the force approached 1.0 N. This shows that the greater the value of the beam thickness, the greater the strain energy, and the more gradual will be the process of strain energy decay over time.

Mathematically,
deflection=f(force, position)

Each of the 3D surfaces shown in Figure 11 is approximated using a polynomial equation as given in Equation (3):(3)$$Deflectionof\;the\;beam\;\left(x=Force,y=position\;at\;any\;point\right)\;=p00+p10x+p01y+p20x^{2}+p11xy$$
where *p*00, *p*10, *p*01, *p*20, and *p*11 are the coefficients of the polynomial.

The results of the coefficients at various beam thicknesses are indicated in Table 6.

By substituting the coefficients in Equation (3), the polynomial equation for each beam thickness can easily be found.

From the above Table 6, it is clear that as the thickness increases, the absolute value of the coefficients decreases, which ultimately reduces the R-squared values. This is the reason why the 3.0 mm thick beam has the lowest values of coefficients and the highest R-squared value. The results can be further simplified in terms of reducing the variables and coefficients. This is performed by plotting the curves of coefficients against beam thickness, which allows the corresponding slopes of the curves (the coefficients w1, w2, and w3) to be determined (see Figure 13).

Each plot in Figure 13 is fitted for a third-degree polynomial, so the generalised equation becomes:(4)$$f\left(x\;thickness\right)=w1x^{3}+w2x^{2}+w3x+w4$$
where *w*1, *w*2, *w*3, and *w*4 indicate the coefficients of the polynomial equation, and *x* indicates the thickness of the beam.

The generalised equation, Equation (4), is simpler to analyse than Equation (3), and is effective in determining the coefficient whatever the thickness of the beam.

The above equations are formulated by substituting the values in the generalised equation. The R-squared value for each coefficient is 1.00, which shows a perfect fit of the curve, as indicated in Table 6. The equations are set for a third-degree polynomial in each case, so there is virtually no discrepancy in the value of any coefficient. This equation can be used to analyse the response of a simple ABS beam of any thickness in the range of 0.5 to 3.0 mm under different loads up to the elastic limit.

### 3.2. Results with the TPU Origami Capsule

In the laboratory, it is possible to design simple experiments in order to examine the deflection of a “cross” TPU capsule held at one tip and with a load applied at the free end; this is effectively a cantilever beam of length 19 mm, width 5 mm and a thickness of 1.0 mm, 2 mm and 3.0 mm. The deflection vs. force curves obtained with the origami capsule for these three thicknesses are shown in Figure 14. The maximum deflection observed for the 3.0 mm thick capsule was 19 mm with an applied force of 1.5 N; the maximum deflection for the 2.0 mm thick capsule was 17 mm for an applied force of 1.3, and the maximum deflection for the 1.0 mm thick capsule was 15 mm for an applied force of 1.0 N.

Using a similar technique to that used with the simple ABS beam, 3D surface graphs for the TPU “cross “were plotted using MATLAB (see Figure 15).

Figure 16 illustrates the response of strain energy with force applied to the TPU “cross” beam. It is clear from the trends that, for all beam thicknesses, the beam’s responsiveness to strain energy is exponential and decreases with the increase in applied force. It is evident that a force of 0.25 N is the maximum force at which all three beam thicknesses showed the response of strain energy.

Deflection as a function of force and position is observed for TPU. From the 3D surface graphs, it is evident that a plastic region was achieved at the maximum values of applied load for each thickness of the TPU “cross” beam.

For the TPU “cross”, the generalised equation for any arbitrary point can be presented by Equation (3).

By substituting the coefficient values in Equation (3), the polynomial equation for each thickness can easily be found.

The R-squared value for each thickness (see Table 6 above) was found to be 0.9966, 0.9818, and 0.9647, respectively, for the 1, 2, and 3.0 mm thicknesses of the “cross” capsule. This indicates that as the thickness of the TPU beam increases, the accuracy of the model equation declines. As with the ABS, the polynomial equation for the TPU “cross” can be simplified using only its thickness. By plotting the corresponding slopes on the curves, the coefficients (w1, w2, and w3) are determined as indicated in Figure 17.

Each graph in Figure 17 was fitted with a second-degree polynomial, so the generalised equation can be presented by Equation (4).

By substituting the values of the coefficient in Equation (4), the generalised equations can be found.

The R-squared values for all four coefficients, as indicated in Table 6 above, were 1.0 for a second-degree polynomial, suggesting a perfect fit for TPU “cross” origami capsule, whereas a perfect fit for the ABS required a third-degree polynomial.

#### 3.2.1. Discussions of Simple ABS Beam and TPU Origami Capsule

The experimental results of the beam were collected and analysed using graphical, simulation, and statistical techniques. The deflection, position, and force vary with thickness, and a 3D gradient graph was plotted in MATLAB to show this (see Figure 11). It is evident from this figure that because the thickness of the beam increased from 0.5 mm to 2.0 mm, the gradient shifted towards a smaller deflection. The 0.5 mm thick beam reached a maximum deflection of 93 mm (see Figure 11a). The 1.0 mm beam achieved a maximum deflection of 90 mm (Figure 11b); the 2.0 mm beam reached a maximum deflection of 78 mm (Figure 11c), and the 3.0 mm thick beam showed a deflection of 64 mm (Figure 11d), were 0.9681, 0.9632, 0.9662, and 0.9819, respectively. This confirms that the thickness of the beam is a critical parameter that moderates the deflection at different values of position and applied force. These values are also significant because they demonstrate that the values of the deflection of the beam can be correlated through a regression model (Table 6).

Of all the results, the beam with the 0.5 mm thickness reported the maximum deflection value, and 3 mm reported the minimum value. This indicates that the thickness of the beam is a critical parameter that modulates the deflection at different values of position and force.

The analysis of the beam’s elastic modulus helped compute the ABS’ resistance to elastic deformation. From the results shown in Figure 10, it can be seen that as the thickness of the beam increased from 0.5 mm to 3.0 mm, the elastic modulus decreased from 3.6 × 109 Pa to 1.8 × 108 Pa. This shows that increasing the thickness of the beam reduces the elastic modulus of the beam. The initiation of the plastic region is indicated by red crosses, as shown in Figure 11. For a 0.5 mm thick beam, the elastic region lasted until the load was 0.108 N, and the plastic region was maintained until 0.549 N. For a beam of 1.0 mm thickness, the elastic region was maintained till 0.343 N, and the plastic region existed up to the maximum load of 1.128 N. For 2.0 mm thickness, the elastic region extended to a load of 1.08 N, significantly more than for the 0.5 mm and 1.0 mm beam thicknesses. Lastly, for a 3.0 mm thick beam, the plastic region was sustained till 4.02 N, showing that for the 3.0 mm thickness, the plastic region commenced at an end load of mass of 310 gm.

The minimum strain calculated to activate the plastic region for the four beams was 0.5 mm—1.79 × 10^−6^; for 1.0 mm—5.70 × 10^−6^; for 2.0 mm—1.74 × 10^−5^; and for 3.0 mm—4.56 × 10^−5^. This shows that as the thickness increased, the plastic region was activated at a greater magnitude of force and a greater overall strain rate. In Figure 12, it is clear that the strain energy is obtained at a maximum of 1 N for all four beams. However, the beam thickness significantly affected the decay of the overall strain energy. This shows that ABS beam may be ideal for low-stress release applications, but for higher stress, the material may not be sufficiently resilient.

In order to simplify the calculation and apply variable thicknesses, the values of all five coefficients from the model equation w00, w10, w01, w20, and w11 were plotted against thickness in Figure 13, and the R-squared values for all four coefficients were 1.0. The R-squared was evaluated for a third-degree polynomial equation but increasing the degree placed the value out of range.

From the experimental results, it is evident that the modulus of ABS increases with strain rate. From the material point of view, the ABS beam depends on both compression and shear rates, which are different for different thicknesses. The elastic limit is reached more rapidly for thin beams and gradually increases as the thickness increases. The reasons for this are the moment of inertia and elastic modulus of the beam, which depend on the properties and cross-sectional dimensions of the material. ABS as a polymer sustained the load to 4.02 N for a 3.0 mm thick beam, indicating that the load sustainability of the designed polymeric beam is suitable for further research work.

The TPU “cross” beam was characterised in order to extend the research work and scope of the study. The analytical process was similar to that for the simple ABS beam, involving beam deflection for different beam thicknesses, in this case from 1.0 mm to 3.0 mm. From an analysis of the 3D curves, as shown in Figure 15, it is evident that the plastic regime had been achieved at the maximum values of applied load for each thickness of the TPU “cross” beam. The R-squared value was 0.9966, 0.9818, and 0.9674, respectively, for the 1.0, 2.0, and 3.0 mm capsule thicknesses. This indicates that as the TPU “cross” origami capsule thickness increases, the accuracy of the model equation decreases, as indicated in Table 6. It can be inferred that the equation is less well-adapted to thicker beams. The maximum stress values recorded for the 1.0, 2.0, and 3.0 mm capsules, respectively, were 6.24 × 10 Pa, 7.61 × 10 Pa, and 3.88 × 10 Pa.

Upon examination of Figure 14, it is evident that the capsule’s elastic limit for the 1.0 mm thick capsule was about 1.04 N, for 2 mm, 1.33 N, and for 3 mm, 1.33 N. Compared to the ABS, the highest elastic modulus was for the 3.0 mm thick beam: 3.88 × 10 Pa, corresponding to the maximum load of 4.02 N. This demonstrates that for the same beam design, the ABS will yield a higher elastic modulus than the TPU “cross”.

In Figure 17, w00, w10, w01, w20, and w11 are plotted against thickness using the model equation; research corresponds to the R-squared values for all coefficients being 1.0. In conclusion, from the experimental results for the TPU “cross” capsule, it is clear that the elastic modulus had a lower value than was achieved with the simple ABS beam. However, one prominent effect that was highlighted for the TPU “cross” capsule was that the beam’s plastic region was activated at relatively low values of applied force. This also suggests high flexibility in the TPU “cross”, which can be used to advantage in those designs where the beam needs to be folded and activated even when there is little change in the applied force. The TPU “cross” is more likely to remain elastic under deformation; this is why we chose to use the TPU “cross” as the material for the capsule and chose ABS to be the beam. It was clear from Figure 16 that the strain energy is obtained at a maximum of 0.25 N for all three capsules. However, the capsule thickness significantly affected the overall decay of strain energy. This suggests that the TPU “cross” capsule may be ideal for higher stresses because the material may be more resilient.

Thus, we inserted a TPU “cross” capsule inside the DCB and calculated the strain energy released by crack propagation in the DCB to assess whether it would activate the TPU “cross”. Since the values of strain had been calculated above, it was easy to pinpoint the amount of strain energy released during crack propagation. The question is whether the amount of strain energy released would activate the TPU “cross” module. The behaviour of the “cross” is observed to be duplicated by the roller under a bending load.

#### 3.2.2. Discussion on Error in Predictions

Once the beam modelling is completed, the validation is performed by adjusting the 3d surface graphs in model approximation and prediction. This enables us to choose the optimum model equation for different materials. As mentioned earlier, the initial process is to assess the model values with the points of the experimental design. The criteria used to test the model fit between different observations and predictions on the deflection, force, and displacement are used. Notably, during the MATLAB plot, the role of determination involved both R2 and adjusted R2, followed by Root Mean Square Error (RMSE).

For the research study, it is questionable what the probable difference between the points obtained from the predicted model versus the experimental design is. Since the number of simulations is not restricted, evaluation of Absolute Error and Root Mean Square Error (RMSE) can be considered for validation. The MSE values obtained for ABS and TPU are indicated in Figure 18 and Figure 19, respectively.

The above-indicated Figure 18 and Figure 19 show the thickness versus error ranges for MSE and RMSE for TPU and ABS, respectively. The highest error range is obtained for ABS, and the lowest is noted for TPU. This is due to the presence of the lowest degree polynomials in ABS model equations and higher in TPU. The MATLAB simulation converts the polynomial equation into an algebraic equation and then carries out the calculations. Therefore, neglecting a higher degree in a calculation in any algebraic equation reduces the model’s accuracy. The only method to reduce error difference is to conduct an experimental research study with precision, as it will reduce analytical and experimental differences.

Theoretically, the proposed numerical model converts the continuous function into a piece-wise function by dividing the domain of the graph into discrete elements. Within this phenomenon, when we try to approximate the continuous function to discrete function, this leads to the generation of error, which generally accounts for a numerical error, in the case of surface graphs, which involve the modelling of the continuous system through discrete elements. With this inherited error, MATLAB does simulate the solution known as a numerically converged solution. However, there also exists a solution that is numerically uncoverged since MATLAB has ignored the inherited error, so the difference between numerically converged and unconverged produces a differential error [44]. Specific to the model equations we proposed, it is presumed the error leading to MSE is the differential error.

### 3.3. Results for Origami “Cross” Module Embedded Structure

For the DCB test, instead of having interlaminar crack growth in the DCB, one of the arms was broken (see Figure 20). The responses of the DCB model can help us to analyse the behaviour of the origami capsule and whether it activates a self-healing mechanism. The analytical process seeks to estimate how much stress is released when the beam is deflected due to the application of a force. With the DCB, it is assumed that the force is dependent on the strain release phenomenon. The response of force vs. displacement for the DCB is presented in Figure 21. Here, the proposed standard was modified by adding a video recorder and camera to the test setup: a picture, which coincided with the force and displacement measurements, was taken every 10 s and was used to visually evaluate and measure the position of the crack tip during the tests with the TPU origami capsule in place.

The load–displacement graphs obtained for the beams with origami capsules and without origami capsules are shown in Figure 21a–d. Nonlinearities in the load–displacement relation were observed for the specimen with the origami capsule. In Figure 21a, the maximum resistant force is 25 N. For the specimen without origami, Figure 21b, it is clear that the maximum force that can be resisted is 19 N at a total displacement of 6.4 mm.

Noticeable is the sudden and substantial drop in force that occurs in both cases, with and without the origami capsule. When the capsule is present, the drop is from about 15 N to 5.4 N, starting at a displacement of about 11 mm. When the capsule is absent, the drop is from about 15 N to 8 N, starting at a displacement of about 8 mm. This sudden failure precedes the full collapse of the DCB. However, the maximum displacement of the beam without the capsule reaches 20 mm, which is significantly higher than that for the beam with the capsule.

We can estimate the percentage error in the experimental deviation using the time and strain released.
(5)$$Strain\;release\;due\;to\;crack\;\left(U\right)=\frac{\sigma^{2}}{2E}\;B\pi a^{2}$$
where σ represents the stress applied, *E* is the elastic modulus of the materials, *B* is the area length, π is the area from middle DCB until the open area, and *a* indicates the length of the beam.

Table 7 indicates the experimental and theoretical values for the strain released (see Equation (5)) during the beam test with and without origami. The results indicate that the difference of strain in the beam with origami is 0.0871, and without origami is 0.0267, which comes to 8.71% and 2.67%, respectively. This shows that the strain released without origami is greater than that with origami. The response of the TPU, which have observed separately, is the same as that of the DCB. In order to use the model equation in calculations relating to self-healing behaviour, we have to include the calculated deviations.

Discussion:

This research assessed the behaviour of origami capsules embedded in the DCB structure. The DCB tests were carried out on an Instron test machine with displacement rates varying from 0.05 to 0.10 mm/s. The energy release rate and values of critical energy were calculated on the basis of the configurations. The material of the specimen was ABS, which was used to evaluate the stress–strain relationship for the DCBs with and without the presence of origami capsules. The results show that the presence of an origami capsule results in a more robust and resilient beam that can withstand greater fluctuations than a beam without an origami capsule. A mathematical analysis of the force vs. displacement and stress vs. strain curves was performed to help assess whether the hypothesis and research arrangement were valid.

From the experimental results, two pairs of graphs were obtained for the beams with origami and without origami, as shown in Figure 21a–d. Figure 21a, shows that with the origami capsule present, the graph proceeds as an almost straight line from 0 N to the maximum force of 24 N, at which the total displacement was 4.8 mm. This denotes the elastic limit of the beam and the resistance at the maximum load. After this point, the beam continued to extend, and the displacement increased, reaching a maximum displacement of 17.5 mm. This was the plastic region of the beam.

For the specimen without origami, as indicated in Figure 21b, it is clear that the maximum force that can be resisted is 19 N at a total displacement of 6.4 mm. The sudden and substantial drop in force from 15N to 5.4 N after this failure was obvious; it was enough to damage the overall beam before it fully collapsed. The maximum displacement of the beam reached 20 mm, which was significantly higher than for the beam with origami.

This is evident from a comparison with the results of previous research work by Simon et al. [40], who carried out load vs. displacement tests on specimens with and without a laminate lay-up. There was a clear difference in delamination lengths for the two specimens, with a rapid drop in load for the non-laminated lay-up compared to a gradual decline in force for the laminated lay-up. This result is similar to our beam results, as shown in Figure 21.

These results indicate that a beam with the origami capsule resists failure better than a beam without the capsule. The beam dimension may also play a significant role in defining the strength of the material. A study by Brunner et al. [42] on the applicability of delamination resistance of different materials indicates that multi-directional lay-ups pose issues due to crack branching and deviation from the plane. The delamination resistance seen in DCB tests depended on the fibre orientation. Alternating the orientations of the cross-ply composites in the beam from 0° to 90° yielded a 50% deviation from the mid-plane.

Figure 21c,d presents graphs of stress and strain, with and without the origami capsule. With the capsule present, the maximum stress was 6 × 106 Pa, and the strain was 0.09, whereas for the specimen without the capsule, the maximum stress was 4.5 × 106 Pa and the strain was 0.11. The stress–strain relation has also been studied by Chen et al. [45] using high-density stitched beams. When the load vs. displacement curve were compared, it was evident that the load increased linearly with the displacement, but when the stitches broke, crack initiation caused a sudden drop in load. Results closer to those in our study are reported by Kato et al. [46]; they found reported an example of crack propagation in DCB made from a satin weave E-glass fabric. The results are comparable in the sense that the delamination was 1.0 mm in width. Additionally, the load fluctuation was also steady, such as that of origami, which indicates the high tensile strength of the material sufficient to bear the increasing load, even during crack formation.

In conclusion, the proposed TPU “cross” origami capsule tends to absorb a sudden fluctuation in load and retards the displacement that may lead to failure. This contrasts with the results of the beam without an origami capsule: these showed a rapid decrease in load with an excessive displacement that led to the failure of the beam. Therefore, the hypothesis that specimens which integrate origami structures are more resilient and exhibit higher tensile strength when loaded axially is supported. Similarly, DCB beams that do not have origami structures embedded within them exhibit lower resilience or lower tensile strength.

### 3.4. Results for a Comparative Study for Strain Energy Activation in an Embedded Structure Versus a Simple TPU Capsule

The results presented in Figure 22 indicate a correlation between time and strain energy for simple TPU capsules 1, 2, and 3.0 mm in thickness. It is seen that the magnitude of the strain energy increases with the thickness of the beam since 3.0 mm showed the highest and 1.0 mm showed the least responsiveness over time.

Figure 23 indicates the strain release over time for the beam with an embedded capsule. It is clear from the trends that the embedded beam of greatest thickness has the highest strain release and vice versa. The shape of the curves was exponential for all thicknesses. Compared to the normal TPU beam structure in which the strain energy lasted for 75 s in the TPU inside an embedded structure, the strain release covered a period of 140 s. The average strain release for 1, 2, and 3 mm beam thickness were 0.00038, 0.01153, 0.0473. Similarly, the standard deviations were 0.00075, 0.00507, and 0.05626 for beams of 1, 2, and 3 mm thickness, respectively.

The generalised equation for any thickness of the beam is represented as:(6)$$y=Strain\;energy=f\left(x=time\right)=ae^{\left(bx\right)}$$
where *y* indicates the strain energy (N.mm), a and b are coefficients, and *x* represents time.

By substituting the values of a and b into Equation (6), the corresponding equation for thickness can be represented.

Figure 24 indicates the responsiveness of the strain release. It is evident that strain release for the DCB with and without the origami capsule has different response times. We see that maximum strain release was attained at 40 s for the beam containing the capsule and 3.5 s for the beam with no capsule. This shows that strain release in the beam with a capsule is higher than for the beam without a capsule.

From the graphs presented in Figure 22 and Figure 23, a third-degree polynomial was used to fit the data points, so the generalised equation becomes:(7)$$f\left(x=time\right)=s1x^{3}+s2x^{2}+s3x+s4$$
where *s*1, *s*2, *s*3, and *s*4 indicate the coefficients of the polynomial equation and *x* indicates the time of the beam (see Table 8).

The R-squared values for with-origami and non-origami capsules are found to be 0.9200 and 0.9333, respectively. This indicates high accuracy for the model equation.

Discussion:

In a structural analysis of the beam’s response to strain energy, it is notable that the beam resists external actions by developing internal stresses induced in the material of the beam by the external forces and their subsequent displacements. The response of these internal stresses also changes with secondary parameters, such as those of geometry and dimensions. In the present research, it was evident that the strain energy for both the embedded structure DCB and the simple TPU beam produces an exponential decay curve. The generalised equation of the curve is indicated in Equation (6), and as shown in Table 9, For both the embedded structure and the simple TPU origami capsule, the value of the constant, a, increases and the value of the exponent, b, decreases with increasing thickness.

Given the similarity in trends for the simple TPU origami capsule and embedded structure, it is evident from Figure 22 and Figure 23 that the magnitude of strain energy increases with the thickness of the beam. In Figure 23, it is observed that the embedded beam with the greatest thickness has the highest strain release, and the converse also holds. Compared to the simple TPU beam structure in which the strain energy lasted for 75 s, the strain release in the TPU-embedded structure extended over a period of 140 s. This shows that strain release was more slowly dissipated in the beam embedded with a TPU capsule than in the simple TPU beam., The response of the 3.0 mm beam is also significant; it shows that the embedded structure can sustain higher strain energy values than the same structure without an embedded capsule. Therefore, when including a healing mechanism in a beam where high strain energy is required, it is necessary to select the thickness of the highest value.

To ascertain the healing rate, we can calculate from Figure 22 and Figure 23 the difference between a TPU in an embedded structure and a simple TPU origami capsule from the extended rate of strain energy. It was found that the embedded structure had an extended dissipation time of 55 secs for all three beams before the trend reached zero. In Figure 24, the time response to the strain release was plotted for beams with and without origami capsule inserts. It is evident that the strain release had a different response time depending on whether an insert was present. For instance, the beams attained maximum strain release at.40 s and 3.5 s, with the larger value corresponding to the beam with the insert, meaning that a crack will not propagate so fast when it is being healed. Because the strain release is dependent on how far the crack has propagated, the strain released due to the presence of the origami capsule acts to resist the crack’s tendency to propagate.

## 4. Conclusions

The research has sought to determine the behaviour of self-healing beams under the elastic and plastic loads for ABS and TPU materials.

The study calculated the strain via strain energy and strain release for beams with and without origami capsules.Origami capsules were made in the shape of a cross with the four small beams comprising the arms, folded or elastically deformed and embedded in the main beam structure.Regarding crack propagation in the main beam, once the strain is released due to the crack, the small beams comprising the origami capsules open in the direction of the crack path and increase the crack resistance of the structure.When ABS transparent was used as the beam and TPU as the embedded capsule, it was found that the capsule worked as a self-healing mechanism, healing the crack before it occurred at a force of 24 N.From the results, it is evident that the properties of the TPU allowed deformation to remain flexible, which is why it was considered a suitable material for a novel self-healing mechanism that can be triggered by crack propagation due to strain release in a structure.Structural analysis has shown that the greater the beam thickness, the greater force required to attain the plastic region.The delamination tests showed the presence of a capsule suppressed the high compressive stresses induced by the bending moment in the vicinity of the crack tip and prevented the specimen from breaking and allowing crack propagation.The results show the potential of origami capsules as a novel self-healing mechanism to extend existing practice, which is primarily based on external healing agents.

## Figures and Tables

**Figure 1 polymers-14-03102-f001:**
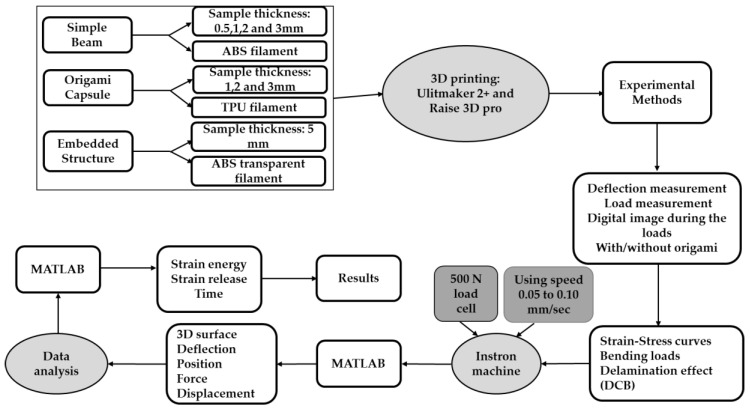
Methodology diagram.

**Figure 2 polymers-14-03102-f002:**
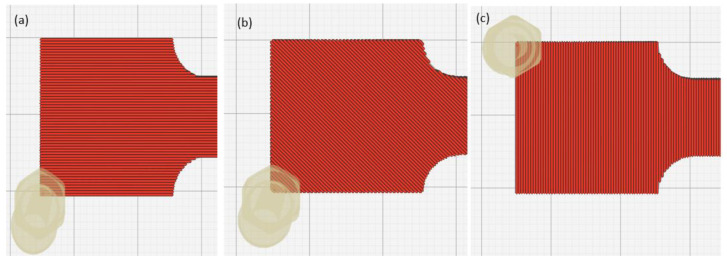
Printing Directions of ABS: (**a**) 0 orientation; (**b**) ±45 orientation; (**c**) and 90 orientation.

**Figure 3 polymers-14-03102-f003:**
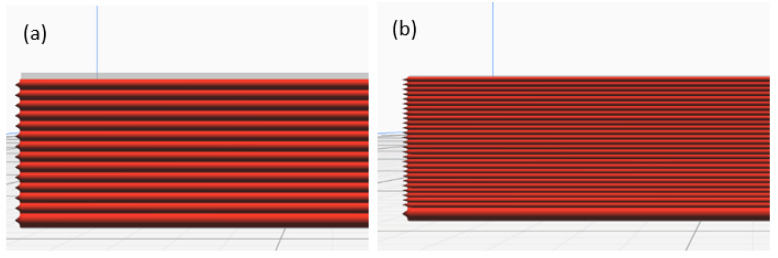
Layer thickness: (**a**) 0.10 mm (**b**) 0.20 mm.

**Figure 4 polymers-14-03102-f004:**
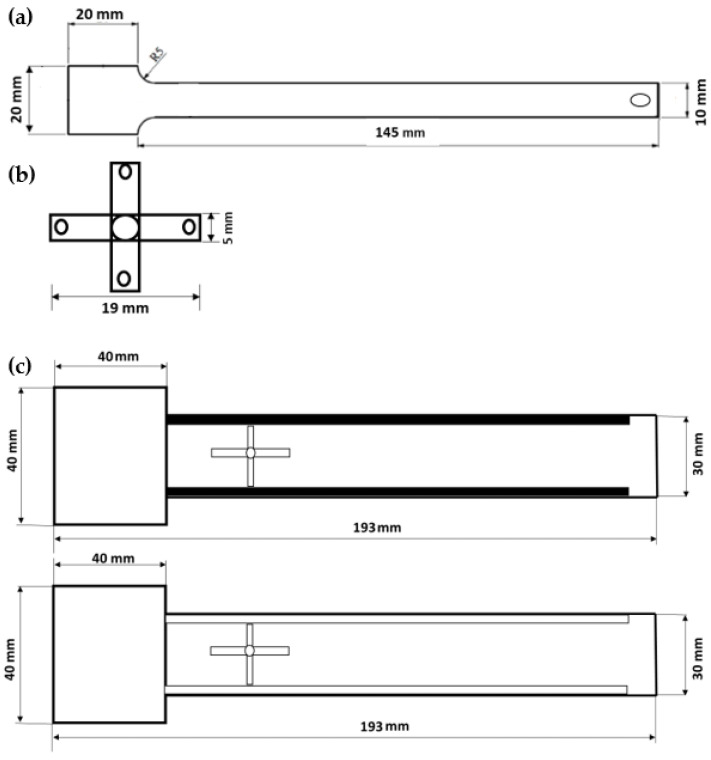
Geometry of specimens: (**a**) ABS simple beam; (**b**) origami capsule TPU; (**c**) Double cantilever beam (DCB (with hole and pillars).

**Figure 5 polymers-14-03102-f005:**
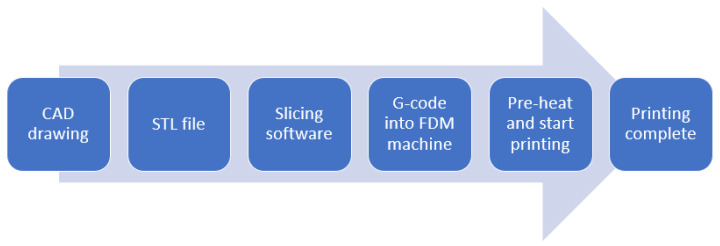
The 3D printer process from drawing to fused deposition.

**Figure 8 polymers-14-03102-f008:**
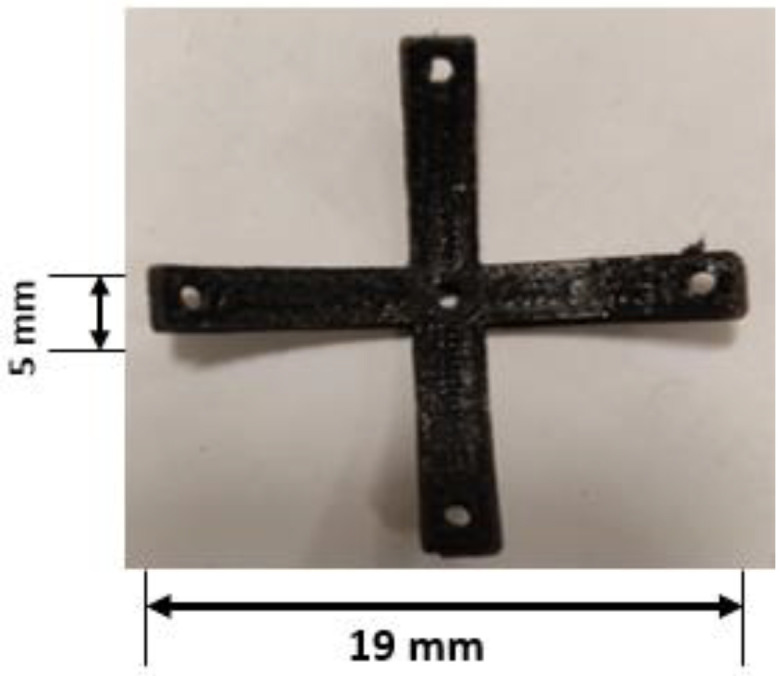
Origami capsule “cross”.

**Figure 9 polymers-14-03102-f009:**
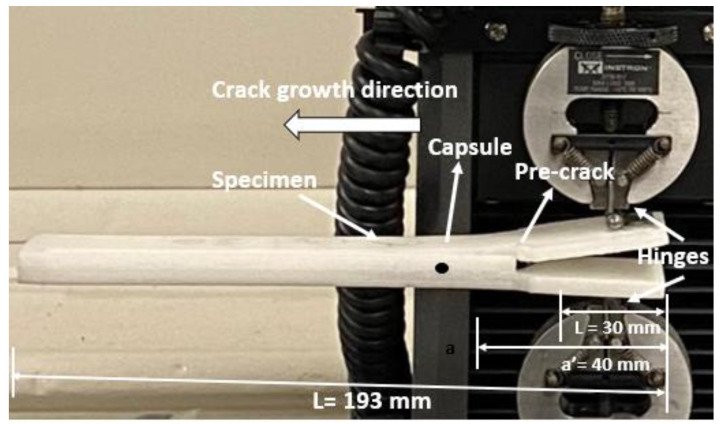
Double cantilever beam (DCB) test setup and showing pre-crack.

**Figure 10 polymers-14-03102-f010:**
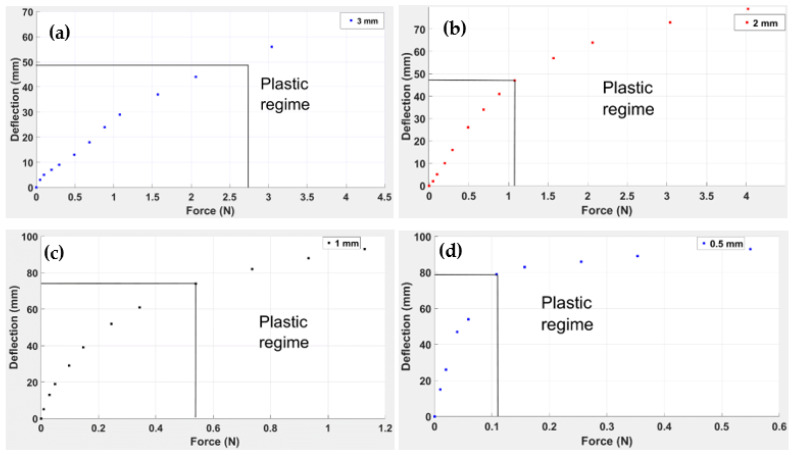
Deflection of end loaded simple ABS beam 145 mm long,10 mm wide and thicknesses: (**a**) 3 mm, (**b**) 2 mm, (**c**) 1 mm and (**d**) 0.5 mm.

**Figure 11 polymers-14-03102-f011:**
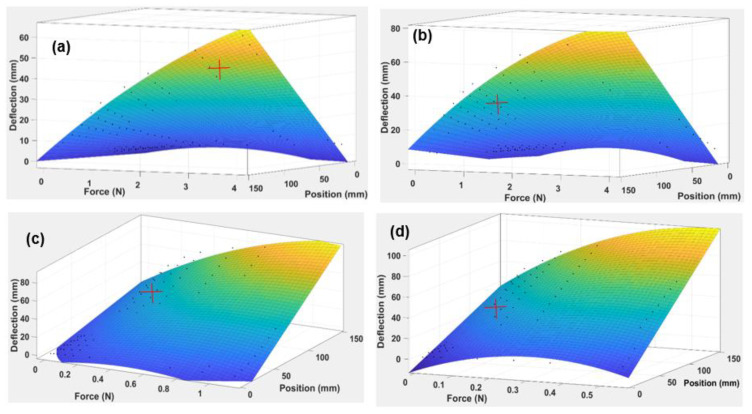
Three-dimensional gradient graph of ABS beam the showing relationship between applied Force (N), Position (mm), and Deflection (mm) for four beam thicknesses: (**a**) 3 mm, (**b**) 2 mm, (**c**) 1 mm, (**d**) 0.5 mm.

**Figure 12 polymers-14-03102-f012:**
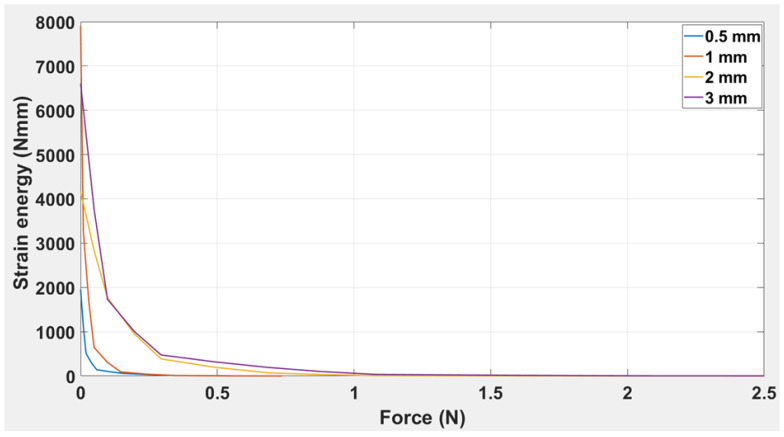
Strain energy vs. applied force for simple ABS beam of thickness: 0.5 mm, 1 mm, 2 mm, and 3 mm.

**Figure 13 polymers-14-03102-f013:**
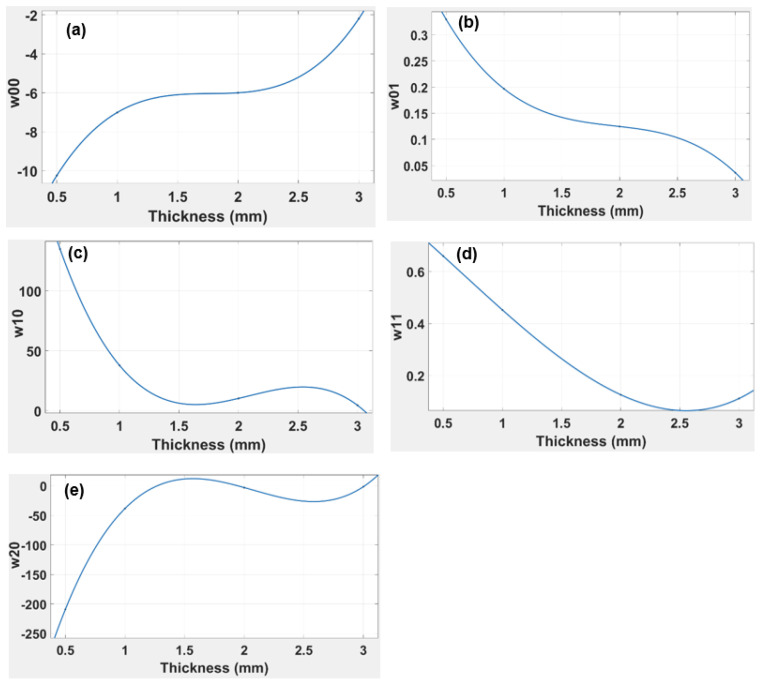
Graphs indicating the relationship of simple ABS beam thickness vs. correlation coefficient for (**a**) w00 (**b**) w01 (**c**) w10 (**d**) w11 and (**e**) w20.

**Figure 14 polymers-14-03102-f014:**
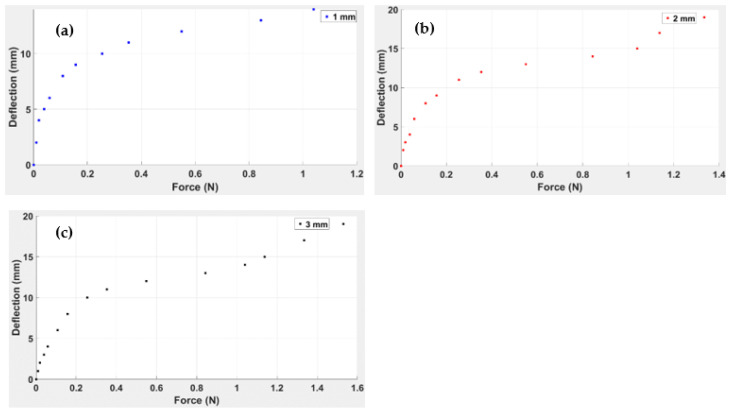
Deflection of an origami TPU “cross” capsule acting as a cantilever beam as a function of force for “capsule” thicknesses: (**a**) 3 mm, (**b**) 2 mm and (**c**) 1 mm.

**Figure 15 polymers-14-03102-f015:**
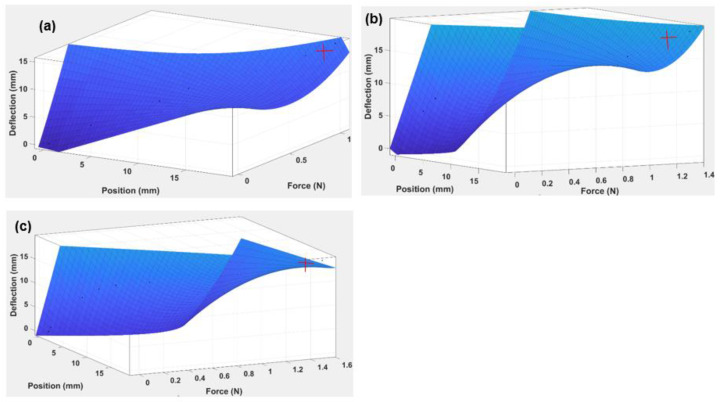
Three-dimensional gradient graph of TPU capsule “cross” showing relationship between Force (N), Position (mm), and Deflection (mm) for three beam thicknesses (**a**) 1.0 mm, (**b**) 2.0 mm, (**c**) 3.0 mm.

**Figure 16 polymers-14-03102-f016:**
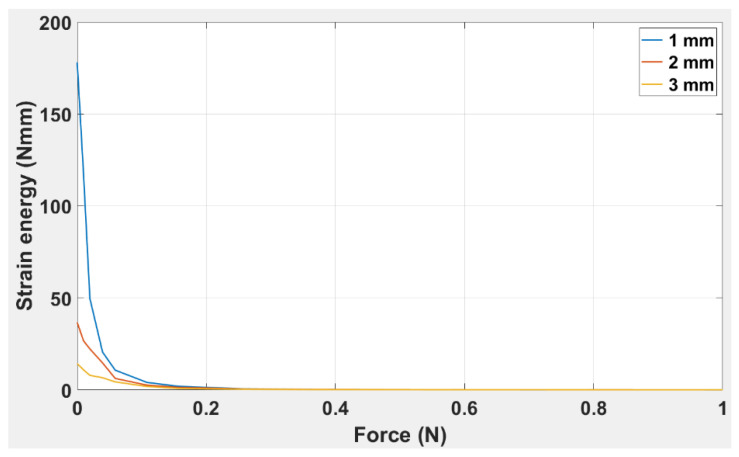
Strain energy versus applied force for TPU “cross” capsule for thicknesses: 1.0 mm, 2.0 mm, and 3.0 mm.

**Figure 17 polymers-14-03102-f017:**
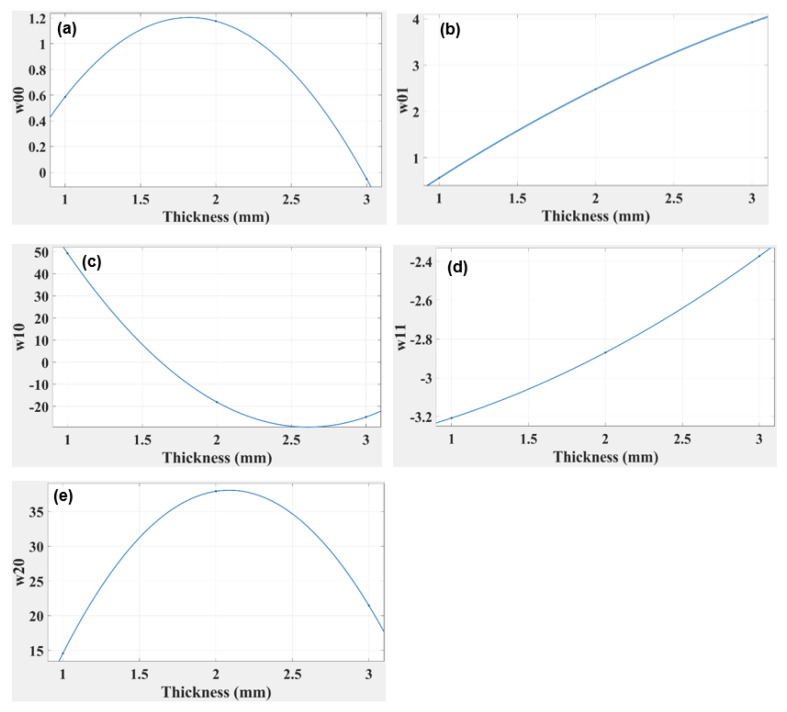
Graphs indicating the relationship of TPU “cross” beam thickness versus the coefficient for (**a**) w00 (**b**) w01 (**c**) w10 (**d**) w11 (**e**) w20.

**Figure 18 polymers-14-03102-f018:**
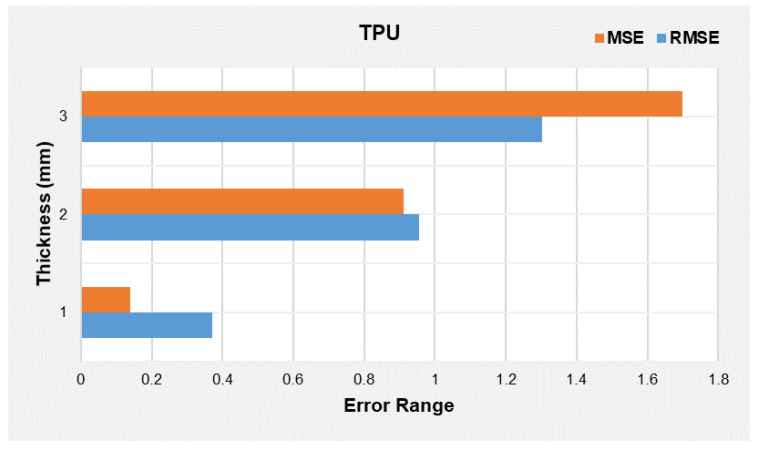
Thickness versus error ranges for MSE and RMSE for TPU beam.

**Figure 19 polymers-14-03102-f019:**
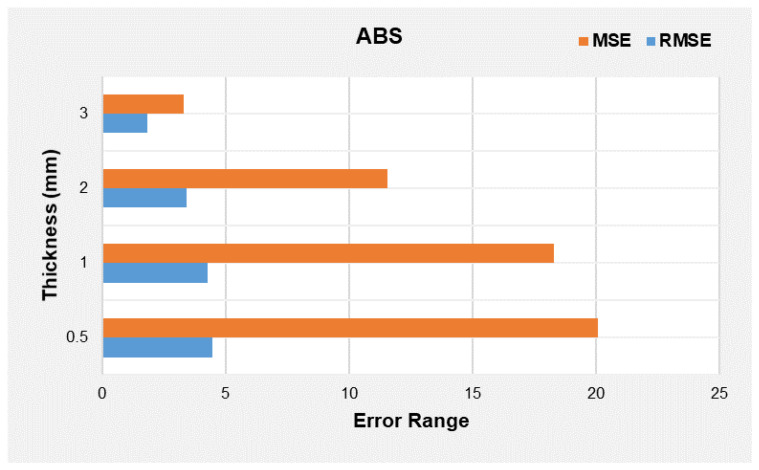
Thickness versus error ranges for MSE and RMSE for ABS beam.

**Figure 20 polymers-14-03102-f020:**
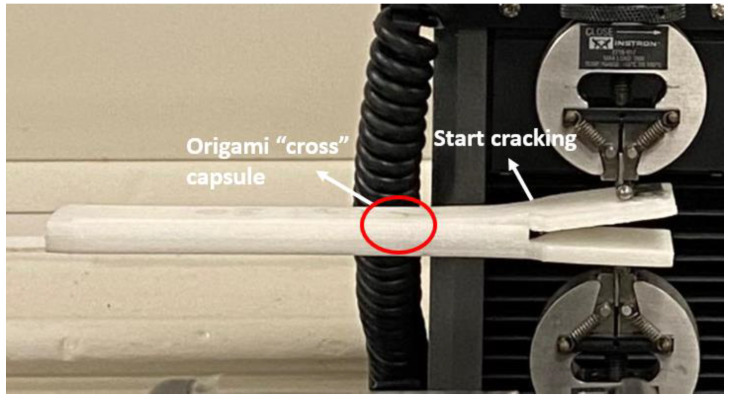
Measuring DCB crack length and displacement during the test.

**Figure 21 polymers-14-03102-f021:**
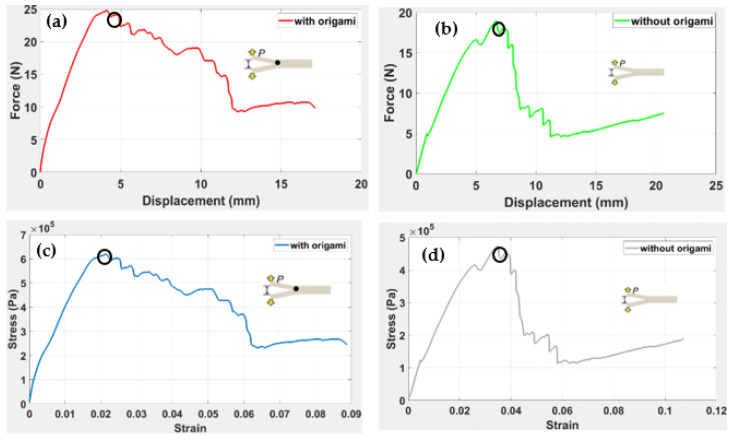
Crack length in DCB under the quasi-static conditions. The load–displacement graphs obtained for the beams with origami capsules and without origami capsules are shown in (**a**–**d**).

**Figure 22 polymers-14-03102-f022:**
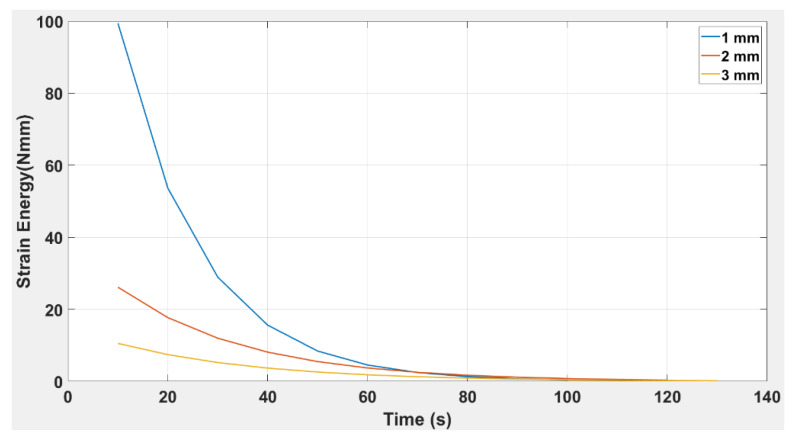
Strain Energy versus time for TPU “cross” beam structure of thicknesses: 1.0 mm, 2.0 mm, and 3.0 mm.

**Figure 23 polymers-14-03102-f023:**
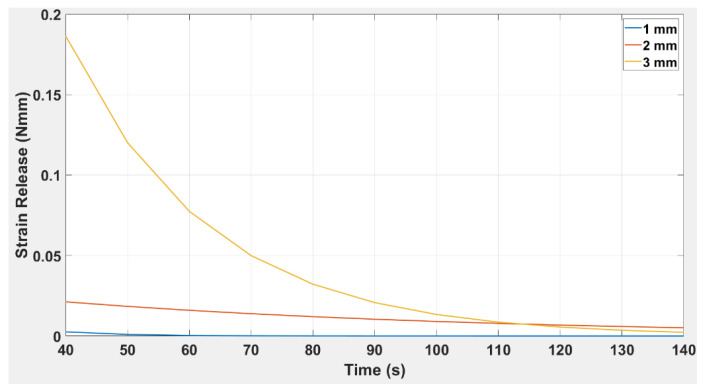
Strain release versus time for TPU inside an embedded DCB beam for thicknesses: 1.0 mm, 2.0 mm, and 3.0 mm.

**Figure 24 polymers-14-03102-f024:**
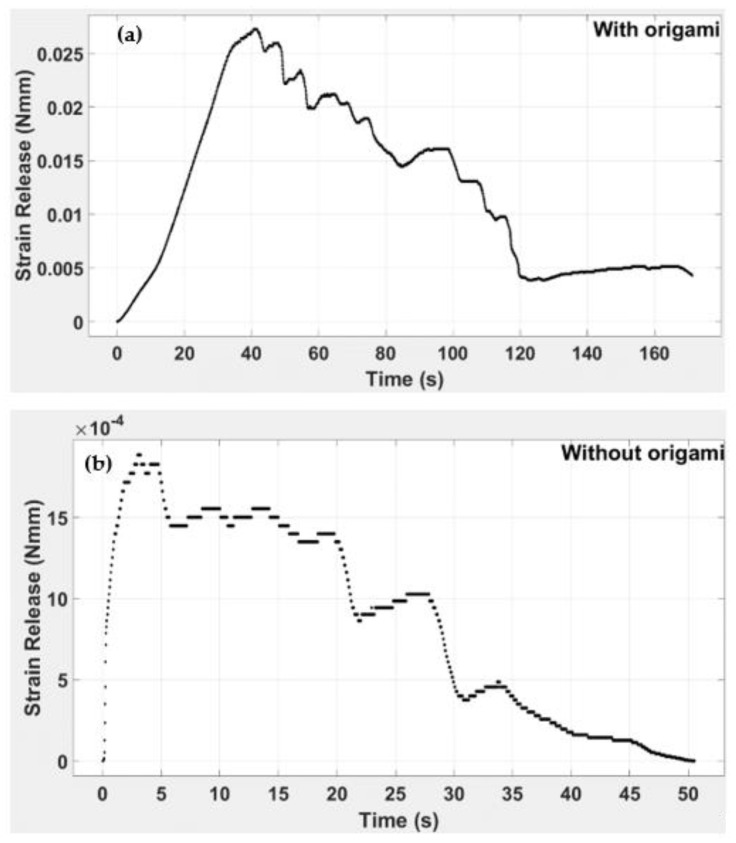
Strain release vs. time for DCB (**a**) with origami capsule and (**b**) without capsule.

**Table 1 polymers-14-03102-t001:** Printing parameters.

Parameters	Value
Nozzle size (mm)	0.4
Layer thickness (mm)	0.1, 0.2
Build orientation	0°, ±45°, 90°
Infill density (%)	100

**Table 3 polymers-14-03102-t003:** Origami capsule designs.

Origami Capsule Shape	Thickness (mm)	Dimensions (mm)	Loads (g)
Cross	1.0, 2.0, 3.0	19L/5W	1, 2, 4, 6, 11, 16, 26, 36, 56, 86, 106

**Table 4 polymers-14-03102-t004:** Experiment Scheme in embedded structure.

Specimen Type	Crack Length	Thickness of Capsule	Mechanical Testing
DCB origami	40 mm	1 mm, 2 mm, 3 mm	Delamination test
DCB without origami	-	-	Delamination test

**Table 5 polymers-14-03102-t005:** Specimen setup.

Specimen	Beam Thickness mm	Dimension, mm (Length/Width)	Type of Structure
with origami capsule	5 mm	193L/30W	With holes and pillars
without origami Capsule	5 mm	193L/30W	With holes and pillars

**Table 6 polymers-14-03102-t006:** Different values of coefficients at simple ABS beams and TPU Origami capsule of different thicknesses.

Coefficients	Simple ABS Beam				TPU Origami Capsule		
	0.5 mm	1 mm	2 mm	3 mm	1 mm	2 mm	3 mm
P00	−10.25	−7.013	−6.002	−2.196	0.5864	1.176	−0.05332
P10	134.8	37.79	10.3	4.519	49.29	−18.06	−24.81
P01	0.3297	0.1968	0.1247	0.0363	0.5673	2.483	3.927
P20	−209.2	−38.41	−2.804	−1.402	14.57	37.92	21.46
P11	0.6598	0.4525	0.1259	0.1112	−3.208	−2.87	−2.373
w00	2.02	−10.70	18.99	−17.32	−0.9095	3.318	−1.822
w01	−0.05	0.32	−0.65	0.58	−0.2359	2.623	−1.82
w10	−40.07	251.30	500.80	327.40	−30.3	−158.3	177.2
w11	0.04	−0.08	−0.37	0.86	0.0795	0.0995	−3.387
w20	74.75	−465.60	909.20	−556.70	−19.91	83.07	−48.59

**Table 7 polymers-14-03102-t007:** Experimental and theoretical model of the beam (a) with origami, (b) without origami.

	with Origami (5 mm)	without Origami (5 mm)
Experimental value of strain release	$\epsilon =8.87\times 10^{-2}$	$\epsilon =2.62\times 10^{-2}$
Theoretical value of strain release	$\epsilon =1.53\times 10^{-3}$	$\epsilon =4.61\times 10^{-6}$
Percent Deviation in strain release	Δϵ = 0.0871 (8.71%)	Δϵ = 0.0267 (2.67%)

**Table 8 polymers-14-03102-t008:** Values of the linear equation coefficients and parametric coefficients for origami and non-origami capsules.

Coefficient	S1	S2	S3	S4	R-Squared
With origami capsule	4.971 × 10^−8^	−1.479 × 10^−5^	0.001144	−0.003213	0.9200
Non origami capsule	5.415 × 10^−8^	−4.429 × 10^−6^	6.075 × 10^−5^	0.001356	0.9333

**Table 9 polymers-14-03102-t009:** Coefficients for DCB with an embedded TPU capsule and simple TPU beam, for three beam thicknesses.

Thickness	Coefficients	Embedded Structure DCB (Average)	TPU Beam Thickness (Average)
1 mm	a	0.1199	184.3
	b	−0.09576	−0.06174
2 mm	a	0.03747	38.61
	b	−0.01419	−0.03903
3 mm	a	1.0784	14.96
	b	−0.04389	−0.03509

## Data Availability

The data presented in this study are available on request from the corresponding author.

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
