# Peer review of "Strain Release Behaviour during Crack Growth of a Polymeric Beam under Elastic Loads for Self-Healing"

_polymers, 2022, doi:10.3390/polym14153102_

Round 1

Reviewer 1 Report

Specific comments:

1."Error! Reference source not found." in various sections. suggest remove the cross referencing field before submitting the manuscript.

2.  suggest providing a flowchart/diagram showing the overview of the project, such as a flowchart, etc.

3. suggest showing a figure showing how the structure is printed, i.e. part orientation.

4. why was the roller origami capsule mentioned even when it was not used in the study?

5. did the author follow any standard when performing the DCB test?

6. figure 8, how was the elastic regime quantified? did the author follow any rule or was any formula used?

7. Graphs presentation requires significant improvement. suggest making the formatting of the graphs consistent.

Author Response

Response to Reviewers

Response to reviewer comments:

We appreciate you and the reviewers for your precious time in reviewing our paper and providing valuable comments. It was your valuable and insightful comments that led to possible improvements in the current version. The authors have carefully considered the comments and tried our best to address every one of them, the amendments are highlighted in red on the revised manuscript. We hope the manuscript after careful revisions meet your high standards. The authors welcome further constructive comments if any.

Reviewer 1:

The authors would like to thank the reviewer for appreciating our work.

Comment -1:

"Error! Reference source not found." in various sections. suggest remove the cross-referencing field before submitting the manuscript.

Reply -1:

Apologies for the mistakes. Due care has been taken in the revised manuscript. Both the mentioned figures and the tables are corrected.

Comment-2:

suggest providing a flowchart/diagram showing the overview of the project, such as a flowchart, etc.

Reply-2:

Authors are agreed with the suggestion. The diagram has been added to the revised manuscript on page 4.

Comment-3:

Suggest showing a figure showing how the structure is printed, i.e. part orientation.

Reply-3:

Authors are agreed with the suggestion. We mentioned two figures in revised manuscript to explain the orientation and layer thickness at figures 2 and 3.

Comment-4:

why was the roller origami capsule mentioned even when it was not used in the study?

Reply-4:

The text is now removed. We have done the testing in roller as well, but not included in the text as the paper seems lengthy. This is the reason we have taken all the discussion out about the roller capsules in the revised version.

Comment-5:

Did the author follow any standard when performing the DCB test?

Reply-5:

Authors are agreed. We have closely followed the steps of ISO 15024. However, the nature of testing of self-healing capsules are unique in their nature and due to this reason we have not referred this standard.

Comment-6:

Figure 8, how was the elastic regime quantified? did the author follow any rule or was any formula used?

Reply-6:

Authors are agreed. We eventually used that value of strain to develop a stress-strain curves and then find out elastic and plastic limits.

A black pen was used to produce speckled dots on the surface at intervals as shown in figure 7. HD camera was used to capture 2D instead. The deflection of each speckled section was measured using the background as reference and inputted into MATLAB.

Comment-7:

Graphs presentation requires significant improvement. suggest making the formatting of the graphs consistent.

Reply-7:

The authors have now improved the graph presentation. The formatting of graphs consistent has now been corrected as suggested.

Reviewer 2 Report

The authors presented a well-written and interesting paper. The following minor comments should be incorporated in the manuscript

1) Concise further your abstract section

2) the paper is quite long, try to combine the discussion sub-sections and present most of the equations as a generic equation with the constants provided in a Table

3) Explain in more depth how would you validate your predictions with some more statistical measures such MSE, ABSE, etc ..

4) Itemize your conclusion section

Author Response

Response to Reviewers 2

Response to reviewer comments:

We appreciate you and the reviewers for your precious time in reviewing our paper and providing valuable comments. It was your valuable and insightful comments that led to possible improvements in the current version. The authors have carefully considered the comments and tried our best to address every one of them, the amendments are highlighted in red on the revised manuscript. We hope the manuscript after careful revisions meet your high standards. The authors welcome further constructive comments if any.

Reviewer 2:

We are very thankful to the reviewer for his comment about the scope of the paper.

Comment-1:

Concise further your abstract section

Reply-1:

Authors are agreed with the suggestion and concise the abstract part in revised manuscript.

Comment-2:

 the paper is quite long, try to combine the discussion sub-sections and present most of the equations as a generic equation with the constants provided in a Table

Reply-2:

The discussion is now more concise as per the suggestion. Generic equations are provided with a table of coefficients. The authors have tried to reduce the length of the paper while keeping the adequate continuity in the discussion.

Comment-3:

Explain in more depth how would you validate your predictions with some more statistical measures such MSE, ABSE, etc.

Reply-3:

Authors are agreed. When we input an equation into MATLAB, the software converts the polynomial equation into a simple algebraic equation for simplification purposes. Due to this conversion, an inherit error is generated. We mentioned it in revised manuscript on page 21and 22.

Once the beam modeling is completed, the validation is performed by adjusting the 3d surface graphs in model approximation and prediction. This enables us to choose the optimum model equation for different materials. As mentioned earlier, the initial process is to assess the model values with the points of the experimental design. The criteria used to test the model fit between different observations and predictions on the deflection, force, and displacement are used. Notably, during the MATLAB plot, the role of determination involved both R2 and adjusted R2, followed by Root Mean Square Error (RMSE).

For the research study, it is questionable what the probable difference between the points obtained from the predicted model versus the experimental design is. Since the number of simulations is not restricted, evaluation of Absolute Error and Root Mean Square Error (RMSE) can be considered for validation. The MSE values obtained for ABS and TPU are indicated in Figures 1, 2, and 3, respectively.

Comment-4:

Itemize your conclusion section

Reply- 4:

Authors are agreed. The conclusion section was itemised as you suggested in revised manuscript.

Round 2

Reviewer 1 Report

The replies are satisfactory.